# Drug Repurposing for Cystic Fibrosis: Identification of Drugs That Induce CFTR-Independent Fluid Secretion in Nasal Organoids

**DOI:** 10.3390/ijms232012657

**Published:** 2022-10-21

**Authors:** Lisa W. Rodenburg, Livia Delpiano, Violeta Railean, Raquel Centeio, Madalena C. Pinto, Shannon M. A. Smits, Isabelle S. van der Windt, Casper F. J. van Hugten, Sam F. B. van Beuningen, Remco N. P. Rodenburg, Cornelis K. van der Ent, Margarida D. Amaral, Karl Kunzelmann, Michael A. Gray, Jeffrey M. Beekman, Gimano D. Amatngalim

**Affiliations:** 1Department of Pediatric Pulmonology, Wilhelmina Children’s Hospital, University Medical Center Utrecht, Utrecht University, Member of ERN-LUNG, 3584 EA Utrecht, The Netherlands; 2Regenerative Medicine Center Utrecht, University Medical Center Utrecht, Utrecht University, 3584 CT Utrecht, The Netherlands; 3Biosciences Institute, Faculty of Medical Sciences, Newcastle University, Newcastle upon Tyne NE2 4HH, UK; 4BioISI-Biosystems and Integrative Sciences Institute, Faculty of Sciences, University of Lisboa, 1749-016 Lisboa, Portugal; 5Physiological Institute, University of Regensburg, D-93053 Regensburg, Germany; 6Centre for Living Technologies, Alliance TU/e, WUR, UU, UMC Utrecht, 3584 CB Utrecht, The Netherlands

**Keywords:** cystic fibrosis, nasal organoids, TMEM16A, screening assay, drug repurposing

## Abstract

Individuals with cystic fibrosis (CF) suffer from severe respiratory disease due to a genetic defect in the cystic fibrosis transmembrane conductance regulator (CFTR) gene, which impairs airway epithelial ion and fluid secretion. New CFTR modulators that restore mutant CFTR function have been recently approved for a large group of people with CF (pwCF), but ~19% of pwCF cannot benefit from CFTR modulators Restoration of epithelial fluid secretion through non-CFTR pathways might be an effective treatment for all pwCF. Here, we developed a medium-throughput 384-well screening assay using nasal CF airway epithelial organoids, with the aim to repurpose FDA-approved drugs as modulators of non-CFTR-dependent epithelial fluid secretion. From a ~1400 FDA-approved drug library, we identified and validated 12 FDA-approved drugs that induced CFTR-independent fluid secretion. Among the hits were several cAMP-mediating drugs, including β2-adrenergic agonists. The hits displayed no effects on chloride conductance measured in the Ussing chamber, and fluid secretion was not affected by TMEM16A, as demonstrated by knockout (KO) experiments in primary nasal epithelial cells. Altogether, our results demonstrate the use of primary nasal airway cells for medium-scale drug screening, target validation with a highly efficient protocol for generating CRISPR-Cas9 KO cells and identification of compounds which induce fluid secretion in a CFTR- and TMEM16A-indepent manner.

## 1. Introduction

Cystic fibrosis (CF) is a monogenic, recessive disease caused by mutations in the gene encoding for the cystic fibrosis transmembrane conductance regulator (CFTR) protein. Currently, 2110 different CFTR gene mutations have been described (http://www.genet.sickkids.on.ca//, accessed on 13 September 2022), which can be further classified based on CFTR defect, i.e., impaired CFTR mRNA/protein expression (class I/VII mutations), protein trafficking (class II) or gating (class III). Depending on the severity of the defect, CFTR dysfunction leads to impaired secretion of chloride and bicarbonate and subsequently affects fluid transport and pH regulation of secreted fluid across epithelial tissues [1,2]. People with CF (pwCF) may therefore experience severe dysfunction of epithelial tissues, including the respiratory tract, pancreas and liver [3].

Respiratory disease in pwCF is mainly caused by CFTR dysfunction in airway epithelial cells, which disturbs airway surface fluid secretion and causes the accumulation of thick mucus in the airways. Thick mucus leads to impaired mucociliary clearance, causing frequent and severe pulmonary infections. Restoring CFTR-dependent fluid secretion in airway epithelial cells may, therefore, reduce respiratory illness by resolving airway mucus obstruction. Recently, CFTR triple modulator therapy (elexacaftor/tezacaftor/ivacaftor), which restores CFTR-dependent anion and fluid secretion in CF airway epithelial cells, has been approved for pwCF carrying at least one copy of the most common F508del class II trafficking mutation [4,5]. Furthermore, CFTR potentiator therapy (ivacaftor) can improve CFTR protein function in pwCF carrying a class III gating mutation [6]. Indeed, current therapeutic strategies to restore CFTR function highly depend on the type of CFTR defect. However, there is a remaining unmet need for ~19% of pwCF who are not eligible for CFTR modulator therapy, and only have access to symptomatic therapies [7]. Importantly, it was reported that among pwCF who are eligible for these drugs worldwide, only 12% are actually having access to them [8], given their excessive cost [9].

Alternative therapies to restore epithelial fluid secretion in pwCF who do not respond to CFTR modulators or do not have access to them may be accomplished in a CFTR-independent manner by activating other chloride channels and transporters. As this approach bypasses CFTR, this CFTR-mutation agnostic approach may be suitable for all pwCF. A promising target for promoting CFTR-independent fluid secretion might be the calcium-activated chloride channel TMEM16A [10,11,12]. TMEM16A was chosen because it is one of the most extensively studied alternative chloride channels and the therapeutic potential is demonstrated by clinical evaluation of the compound ETD002, which enhances activity of TMEM16A in pwCF [13]. An alternative approach to modify airway epithelial fluid homeostasis is through modulation of the epithelial sodium channel ENaC [14,15]. However, clinical interventions targeting ENaC have thus far not yielded clear clinical benefit [14,15,16]. A large effort in the field currently aims to identify and prioritize additional therapeutic targets that modify airway epithelial ion and fluid secretion, independently of CFTR. However, the study of such pathways and targets in primary airway epithelial cells remains technically highly challenging due to lack of sufficient throughput in assays and the inability to efficiently genetically engineer primary airway cells.

In a previous study, we described a method that enables culturing of airway organoids from minimal-invasive nasal brushings [17]. Nasal organoids resembled a mucociliary differentiated airway epithelium. Furthermore, organoid swelling was used to measure epithelial fluid secretion induced by CFTR-modulating drugs, but we also observed CFTR-independent fluid secretion in organoids from pwCF. Based on this, we proposed that CF nasal organoids can be used as a platform to identify mechanisms of CFTR-independent fluid secretion and to identify modulators of such pathways.

Here, we miniaturized the previously described airway organoid fluid secretion assay to a 384-well plate format to facilitate primary airway epithelial fluid secretion studies at higher throughput. This assay was then used to screen ~1400 FDA-approved drugs for agonists of fluid secretion in CF primary nasal airway organoids. A brightfield image analysis platform was further developed to measure nasal organoid swelling as result of luminal organoid fluid secretion, based on a previously developed artificial intelligence-based imaging platform [18]. Finally, we demonstrate an efficient procedure to generate a gene knockout (KO) in primary nasal epithelium for mode-of-action studies of validated hits.

## 2. Results

### 2.1. Nasal Organoids from Donors without Functional CFTR Display CFTR-Independent Fluid Secretion

First, we wanted to confirm that nasal organoids can be used to measure CFTR-independent fluid secretion as suggested by previous work with F508del/F508del nasal organoids [17]. Cryopreserved airway basal progenitor cells were fully differentiated at an air–liquid interface (ALI), followed by formation of nasal organoids from epithelial sheets of these differentiated ALI cultures (Figure 1A). These organoids (F508del/S1251N) were well-differentiated, displaying both MUC5AC+ goblet cells and β-tubulin IV+ ciliated cells (Figure 1B). In line with earlier observations in CF nasal and bronchial organoids [17,19], nasal organoids from a donor without CFTR function (1811+1G>C/1811+1G>C, a severe splice mutation) showed intrinsic lumen formation without any stimulation (Figure 1C), suggesting CFTR-independent epithelial fluid transport. To demonstrate detectable expression of alternative ion channels and transporters, we conducted qPCR experiments in nasal organoids from three donors with class I/VII mutations, leading to no CFTR protein (W1282X/1717-1G>A, R553X/R553X, G542X/CFTRdele2.3 (21 kb)). Here we confirmed detectable mRNA expression of ANO1/TMEM16A, SLC26A9, CLCN2 and ENaC/SCNN1A (Figure 1D). Interestingly, mRNA expression of some ion channels or transporters is higher compared to others; however, further research is needed in a larger donor cohort to demonstrate a relationship between CF disease and alternative ion channel expression. The same organoids displayed CFTR-independent forskolin-induced swelling (FIS) as quantified by 2h organoid swelling measurements (Figure 1E–G). Furthermore, organoid swelling was also observed after stimulation with ATP or Eact, potentially through modulation of calcium-dependent channels or transporters, including TMEM16A and TRPV4 (Figure 1E–G) [20]. Altogether, these findings confirm CFTR-independent fluid secretion in CF nasal organoids.

### 2.2. Nasal Organoid Swelling in a 384-Well Plate Format

We next set out to develop a 384-well plate fluid secretion screening assay in nasal organoids to enable analysis of ion and fluid transport studies at higher throughput. As calcein green labeling of nasal organoids was technically challenging in this plate format, we developed an alternative approach for quantifying organoid swelling based on organoid recognition with OrgaQuant [18]. This is an open-source deep convolutional neural network, trained to recognize cystic intestinal organoids in brightfield images. To validate organoid recognition and quantification of organoid swelling in 384-well plates using OrgaQuant, we determined the effect of the CFTR potentiators VX-770 (ivacaftor) and PTI-808 (dirocaftor), which both facilitate opening of the CFTR channel, on FIS in CF nasal organoids with a S1251N CFTR gating mutation. Assay validation was performed in this particular donor with high responses to CFTR-modulating drugs to obtain a larger range of organoid swelling measurements. The OrgaQuant model was indeed able to recognize spherical nasal organoids, and the surface area could be estimated using the OrgaQuant bounding boxes, assuming organoids had a disk shape (Figure 2A). Moreover, based on particle tracking, we were able to follow individual organoids over time by live-imaging [21]. Organoid surface areas were subsequently calculated for each time point, linear regression was performed to determine swell rates for individual organoids and the mean swell rate of all organoids within a well was calculated (Figure 2B). Based on this quantification method, we observed a significant increase in FIS of organoids stimulated with VX-770 or PTI-808 compared to vehicle control or forskolin alone (Figure 2C). Furthermore, we observed a correlation between swell rates that were measured with the OrgaQuant model and the conventional quantification method with area under the curve (AUC) values from fluorescent-labeled organoids (Figure 2D,E). In addition to CFTR modulator responses in FIS, we were able to quantify CFTR-independent Eact-induced swelling in CF nasal organoids in brightfield images using the OrgaQuant model (Figure 2F). Thus, we conclude that this newly developed method to quantify organoid swelling in brightfield images is suitable for fluid secretion assays in a 384-well plate format.

### 2.3. Screening of FDA-Approved Drugs in CF Nasal Organoids

Next, we aimed to identify FDA-approved drugs that could induce CFTR-independent fluid secretion in CF nasal organoids. We conducted a primary screening assay in which we examined the effect of ~1400 FDA-approved drugs on organoid swell rate over 3 h using nasal organoids from four pwCF (F508del/F508del, F508del/F508del, F508del/W846X, 1811+1G>C/1811+1G>C; Figure 3A). These donors had no functional CFTR at the plasma membrane, due to mutations leading to impaired trafficking, misfolded CFTR or no production of CFTR mRNA or protein. Two compounds were combined within a single well of a 384-well plate to reduce experimental conditions and time, and Eact, a compound known to activate calcium-dependent fluid secretion, was used as a positive control. Because of the variation in baseline swelling (without compound) among assay plates and donors (Appendix A), a plate-normalization step was performed (Appendix A). After normalization, swell rates from different replicates and donors were averaged for all compounds (Figure 3B). Wells with a plate-normalized swell rate higher than 1, corresponding to 1 interquartile range (IQR) above the median swell rate of all compounds, were defined as a hit (Figure 3B–E). This resulted in 90 compounds, divided over 45 wells, that were selected for validation in a secondary screening.

### 2.4. Identification of Hit Compounds That Induce CFTR-Independent Organoid Swelling

To further identify and validate the hit FDA-approved compounds from the primary screening assay on their potential to induce CFTR-independent fluid secretion, we conducted a secondary screening assay. This was executed using the conventional 96-well plate format. Secondary screening was conducted in the same four CF donors as the primary screening assay, which lack functional CFTR at the plasma membrane, but now testing only one compound per well (Figure 4A and Appendix A). Amongst the compounds that were screened in the secondary screening assay, 12 hit compounds were selected for further studies (Figure 4B,C). Selection was based on a combination of safety profile, experience with chronic use in other diseases and effectiveness in the CFTR-null donor (1811+1G>C/1811+1G>C). Additionally, a selection of compounds with a similar mode-of-action was made, as many cAMP-inducing compounds, including ß2-adrenergic agonists, were among the top compounds. Next, to exclude a potential role for residual CFTR function in mediating nasal organoid swelling by the hit compounds, we conducted a final validation study in nasal organoids from three other CFTR-null donors (W1282X/1717-1G>A, R553X/R553X, G542X/CFTRdele2.3 (21 kb)). These experiments were performed with the conventional swelling assay with fluorescent-labeled organoids in 96-well plate format. All compounds, except Benidipine, significantly induced swelling of nasal organoids from CFTR-null donors (Figure 4D–F). The discrepancy in significance between compounds with a similar effect size might be due to a higher number of measurements for some compounds, resulting in higher power for statistics. Differences in the magnitude of organoid swelling and ranking of the hit compounds between the secondary screening assay (Figure 4B) and the validation in CFTR null donors (Figure 4D) might be due to the use of different donors or different compound batches (FDA drug library versus powders).

### 2.5. Generation and Validation of TMEM16A KO Nasal Epithelial Cells

Based on earlier results in swelling assays with the TMEM16A activator ATP (Figure 1E,F), we hypothesized that TMEM16A may contribute to CFTR-independent nasal or-ganoid swelling. To further investigate this, we created TMEM16A gene KO nasal epithelial cells using the CRISPR-Cas9 technology. These TMEM16A KO cells were generated by use of electroporation in nasal cells from three CFTR-null donors (W1282X/1717-1G>A, R553X/R553X, G542X/CFTRdele2.3 (21 kb)). To enhance efficiency, the TMEM16A locus was targeted using a mix of three different sgRNAs (Figure 5A). KO efficiency was first validated by DNA gel electrophoresis (Figure 5B) and Sanger sequencing, which revealed KO efficiencies of 95%, 87% and 88% for the three donors, respectively. Because of the high KO efficiencies, a selection step was not needed, and we therefore continued with a polyclonal cell population. For optimal validation of the TMEM16A gene KO efficiency, we also characterized cells that were stimulated with the pro-inflammatory cytokine IL-4, which is known to enhance TMEM16A expression and function [10,22]. First, TMEM16A protein levels were assessed in ALI-differentiated KO cells using Western blot under normal and IL-4 treated conditions (for 48 h). Under both conditions, a near-complete KO phenotype was observed (Figure 5C,D). As a final functional validation, TMEM16A-dependent chloride conductance was determined in ALI-differentiated KO nasal cells in Ussing chamber experiments (Figure 5E,F). UTP-induced short-circuit currents (Isc), which were sensitive to the TMEM16A-inhibitor Ani9, were used as a measurement of TMEM16A activity [23]. Quantification of the UTP-induced peak currents showed a partial Ani9-sensitivity in the control cells, suggesting TMEM16A to be responsible for more than half of the UTP-induced currents. In TMEM16A KO cells, UTP-induced currents were significantly reduced and not affected by Ani9. As expected, in IL-4-treated control cells, UTP-stimulated currents were markedly enhanced and inhibited significantly by Ani9. In IL-4-treated KO cells, UTP-stimulated currents were small, and not affected by Ani9, similar to non-IL-4 KO cells. Altogether, these results confirm a functional TMEM16A KO in CFTR-null nasal epithelial cells, which can be further used to examine whether TMEM16A mediates nasal organoid swelling.

### 2.6. Hit Compounds Induce TMEM16A-Independent Fluid Secretion

Next, we aimed to determine whether the 12 remaining hit compounds induced swelling of organoids, generated from ALI-differentiated TMEM16A KO nasal cells. We observed no differences in lumen formation between TMEM16A KO and control organoids without any stimulation (Figure 6A,B). This suggests no role of TMEM16A in intrinsic lumen formation of CFTR-null nasal organoids. Next, ATP-induced organoid swelling was studied, as ATP is known to activate TMEM16A. We did not observe a decline in ATP-induced swelling in TMEM16A KO organoids in regular organoid culture conditions. In contrast, ATP-induced organoid swelling was reduced in IL-4-treated TMEM16A KO organoids (Appendix A). This suggests that ATP-induced swelling mediated by TMEM16A is only detected upon IL-4-stimulation. Next, we further validated the 12 FDA-approved hit compounds in TMEM16A KO nasal organoids. However, no significant differences were observed in swelling between KO and control organoids, suggesting no TMEM16A involvement in epithelial fluid secretion induced by the hit compounds (Figure 6C–E).

### 2.7. Effect of the Hit Compounds on Chloride Conductance and TMEM16A Activating Effects

We further conducted mode-of-action studies with the hit compounds to determine effects on chloride conductance and TMEM16A-activating effects. First, the effect of acute addition of the hits on resting Isc was investigated in ALI-differentiated CFTR-null nasal cells (W1282X/1717-1G>A, R553X/R553X, G542X/CFTRdele2.3 (21 kb)) by Ussing chamber measurements (Figure 7A). In contrast to organoid swelling, the hit compounds did not induce any measurable change in Isc in ALI cultures of corresponding donors (compare Figure 7A to Figure 4D). We then investigated if the hit compounds had any stimulating effect on TMEM16A-dependent chloride transport mediated by TMEM16A agonists, i.e., UTP, ATP and ionomycin. We first examined the effect of the hit compounds on the response to a range of UTP concentrations (0.1 to 100 µM) in fully differentiated CFTR-null nasal epithelial cells with Ussing chamber measurements. However, none of the compounds enhanced these UTP-induced currents (Figure 7B and Appendix A). In addition to ALI-differentiated CFTR-null nasal epithelial cells, TMEM16A-dependent chloride transport was studied in YFP-quenching assays in two different cell lines. The hit compounds did not enhance Ani9-sensitive YFP-quenching upon Ionomycin stimulation of CFBE cells (Figure 7C,D and Appendix A), nor upon ATP stimulation in a HT-29 cell line (Figure 7E,F). Altogether, these experiments in other in vitro model systems indicate no effect of the hit compounds on chloride conductance measured in the Ussing chamber, and the lack of additional stimulating effect on TMEM16A-dependent chloride transport.

## 3. Discussion

In this study, we performed a screening assay in CF nasal organoids, with the aim to repurpose FDA-approved drugs that stimulate CFTR-independent fluid secretion. Screening assays in 384-well plate format using airway organoids have been previously been described by others [24,25,26] and our protocol was based on a 384-well screening assay for CFTR-modulating drugs in CF intestinal organoids [27]. However, to our knowledge, this is the first medium-throughput 384-well screening assay using nasal airway organoids that are cultured from minimal invasive nasal brushings of pwCF. In contrast to previously described assays using airway epithelial cells derived from resected tissues, the use of nasal organoids enables personalized disease modeling in airway cells of pwCF with any CFTR genotype.

Of the ~1400 FDA-approved drugs, 12 hit compounds were identified to induce fluid secretion in CFTR-null nasal organoids, based on assessment of organoid swelling. To exclude a role of CFTR in nasal organoid swelling, the hit compounds were subsequently tested in CFTR null nasal organoids. Ideally, the primary and secondary screening assays would already have been performed in these CFTR null nasal organoids, but these cells were not yet available at the beginning of the study. The screening assays were, therefore, performed on some donors with a trafficking mutation. It can be speculated that the identified compounds induce organoid swelling by restoring CFTR function. However, we assume this is highly unlikely based on the 2 h compound incubation period, which is too short to observe CFTR functional repair caused by enhanced protein expression or trafficking.

To study the possible role of the alternative chloride channel TMEM16A in CFTR-independent nasal organoid swelling, CRISPR/Cas9 was used to create a TMEM16A KO in CFTR-null patient-derived nasal epithelial cells. We achieved high KO efficiencies (87–95%) and a selection step for clonal expansion was, therefore, not required. Despite functional validation of impaired TMEM16A function in nasal epithelial cells, nasal organoid swelling induced by the hit compounds was not reduced in TMEM16A KO cells. Further mode-of-action studies in different in vitro models indicated no direct effect of the hit compounds on transepithelial ion transport measured in the Ussing chamber, nor any stimulating effect on TMEM16A-dependent chloride transport, suggesting a TMEM16A-independent mode-of-action of the hit compounds. In line with previous studies [10,22], we observed that IL-4 could boost TMEM16A expression and TMEM16A-dependent swelling of CF nasal organoids in response to ATP. Therefore, nasal organoid culture conditions with IL-4 can potentially be used to more specifically identify TMEM16A activating compounds in future screening assays.

The discrepancy which was observed between the effects of the hit compounds on organoid swelling and Ussing chamber measurements can have multiple causes. In the Ussing chamber, changes in transepithelial ion transport are directly measured as electric currents, while organoid swelling is an indirect measurement of ion transport. Therefore, in contrast to electrical currents measured in the Ussing chamber, fluid secretion in organoids might depend on non-electrogenic transporters, e.g., by the electroneutral Cl^−^/HCO_3_^−^ exchanger pendrin (SLC26A4) [28,29]. Moreover, ALI-cultured monolayers used in the Ussing chamber might display differences in the activity of ion channels and transporters compared to organoids. For instance, the biomechanical properties of the 3D-extracellular matrix in which organoids are cultured may affect the activity of mechano-sensitive ion channels, such as TRPV4 that is activated by Eact [20,30]. This would correspond with observations made in ALI cultures under shear stress, in which CFTR-independent chloride conductance and fluid secretion are also observed [31]. Additionally, fluid secretion experiments are cumulative assays that measure ion transport over a 2–3 h time period and might, therefore, be somewhat more sensitive to detect low signals. To further compare the differences between organoid and ALI cultures, airway surface liquid depth measurements and pH regulation studies of ALI-differentiated cells may be performed to examine fluid transport induced by the hit compounds.

The selection of hit compounds included many cAMP-inducing agents. Besides CFTR [32] and also TMEM16A [33], it has been described that cAMP can modulate the activity of other ion channels and transporters. For instance, cAMP agonists activate the Cl^−^ channel CLCN2 [34], SLC26A9 [35], the Cl^−^/HCO_3_^−^ exchanger pendrin (SLC26A4) [36], the Na^+^/HCO_3_^−^ cotransporter NBCE1 [37] and the H^+^/K^+^ ATPase HKA2 (ATP12A) [29,38]. Additionally, it has been shown that cAMP modulates the expression of the Cl^−^/HCO_3_^−^ exchanger type 2 (AE2 or SLC4A2) in airway epithelial cells [29,39,40]. Furthermore, cAMP mediates K^+^ signaling, e.g., by activation of the basolateral membrane K^+^ channel KCNQ1 [39,41], which is associated with chloride secretion [42]. Likewise, cAMP is reported to affect Ca^2+^ signaling [43] and, therefore, might activate a non-TMEM16A calcium-activated chloride channel [44]. Lastly, cAMP is described to mediate Na^+^ transport via the epithelial sodium channel ENaC [45]. Altogether, the mechanism-of-action of cAMP-enhancing drugs might depend on the interplay among different ion channels and transporters, which should be investigated in further research. Transcriptomics and proteomics can be performed to identify ion channels or transporters that are highly expressed in nasal organoids of individuals with CF. This might be followed by the creation of gene KO cells of these specific ion channels and transporters, to validate their involvement in CFTR-independent fluid secretion. In addition, studies can be conducted with chemical inhibitors to elucidate which cellular signaling transduction pathways are activated by the hit compounds.

Among the cAMP-stimulating hit compounds are multiple β2-agonists, such as Terbutaline Sulfate, Salbutamol and Indacaterol Maleate. They seem attractive for drug repurposing as they are broadly applied as bronchodilator therapy for respiratory diseases. Indeed, a significant number of pwCF already use bronchodilator inhalation therapy to reduce respiratory symptoms of airway obstruction [46]. Notably, these β2-agonists are also described to enhance CFTR-dependent epithelial permeability in human bronchial epithelial cells [47] and to induce CFTR-dependent fluid secretion in intestinal organoids [48]. As we found that they also induce non-CFTR epithelial fluid secretion, they might have a double positive effect for pwCF. Among the hit compounds, we also observed that the β2-adrenergic antagonist Labetalol induced organoid swelling. This is potentially by acting as a partial agonist of the β-adrenoceptor, as previously shown by others [49]. However, further research is needed to fully understand the working mechanism of Labetalol and other hit compounds before considering clinical use, as it is preferable to have drugs that activate specific chloride channels or transporters to prevent systemic side effects. After elucidating the working mechanisms, further studies can also be conducted to determine additive or synergistic effects of compounds with different mode of actions.To further explore the therapeutic potential, additional studies may include comparison of the effect sizes of the hit compounds with current CFTR modulators, by determining swelling of organoids from patients that respond to CFTR modulator therapies. Moreover, the effect of the hit compounds can be determined as add-on therapy together with CFTR modulators. 

In summary, we provide proof-of concept of using nasal organoids for medium-throughput screening and the ability to elucidate the mechanism-of-action of hit compounds in gene KO cells. We furthermore identified 12 FDA-approved compounds which induce CFTR- and TMEM16A-independent epithelial fluid secretion in CF nasal organoids and may potentially be used as treatment for pwCF. Moreover, our pipeline, combining screening assays in nasal organoids and validation experiments in gene KO cells can be further used for pre-clinical drug discovery. It can be used to identify novel compounds that activate alternative ion channels or transporters, which might act as treatment for pwCF who are not eligible for CFTR modulator therapy.

## 4. Materials and Methods

### 4.1. Patient Materials

Nasal brushings were obtained from subjects with CF. All subjects signed informed consent for use and storage of their cells, which was approved by a specific ethical board for the use of biobanked materials TcBIO (Toetsingscommissie Biobanks), an institutional Medical Research Ethics Committee of the University Medical Center Utrecht (protocol ID: 16/586). Cells were used from 9 pwCF with the following mutations: F508del/S1251N (Female (F)); F508del/S1251N (Male (M)); W1282X/1717-1G>A (F); R553X/R553X (F); G542X/Dele2.3 (21kb) (M); F508del/W846X (F); F508del/F508del (M); F508del/F508del (F); 1811+1G>C/1811+1G>C (F). Nasal brushings were performed by a trained nurse or physician, as described before [17]. Briefly, the brushings were obtained from both inferior turbinates using a cytological brush (CooperSurgical, Trumbull, CT, USA) and were collected in advanced DMEM/F12 (Gibco, Waltham, MA, USA) containing glutaMAX (1% *v*/*v*; Gibco, Waltham, MA, USA), HEPES (10 mM; Gibco, Waltham, MA, USA), penicillin-streptomycin (1% *v*/*v*; Gibco, Waltham, MA, USA) and primocin (50 mg/mL; Invivogen, San Diego, CA, USA).

### 4.2. Nasal Epithelial Cell and Organoid Culturing

Nasal epithelial cells were isolated and expanded as described previously [17]. In brief, cells were scraped off the brush and treated with TrypLE express enzyme (Fisher Scientific, Landsmeer, The Netherlands), supplemented with Sputolysin (Calbiochem, San Diego, CA, USA) for 10 min at 37 °C. Subsequently, cells were strained with a 100 µM strainer, centrifugated and plated out in a collagen IV (50 µg/mL; Sigma-Aldrich, St. Louis, MO, USA)-precoated 6-well culturing plate with basal cell isolation medium (Appendix A). Growth factors (FGF7, FGF10, EGF and HGF) were added fresh to the medium. Medium was changed three times a week. Antibiotics were withdrawn from the medium after one week of culturing and cells were further cultured with basal cell expansion medium, including the γ-secretase inhibitor DAPT (Appendix A) until 80–90% confluence. Cells were then passaged using TrypLE express enzyme and further expanded until confluence. These cells were frozen as a master cell bank (passage 1) and working cell bank (passage 2) in CryoStor CS10 freezer medium (STEMCELL technologies, Vancouver, Canada), supplemented with Y-27632 (5 µM; Selleck chemicals, Planegg, Germany). For experiments, basal cells (passage 3–5) were seeded on 12-well inserts (0.4 μm pore size polyester membrane, 0.5 million cells per transwell; Corning, Corning, NY, USA), precoated with PureCol (30 ug/mL; Advanced BioMatrix, Carlsbad, CA, USA) for differentiation at air-exposed conditions. The cells were first cultured submerged with basal cell expansion medium. When reaching 100% confluence, medium was changed to ALI differentiation medium (Appendix A) supplemented with A83-01 (500 nM). After 2 days, apical medium was removed to culture the cells at air-exposed conditions. After 3–4 days at air-exposed conditions, A83-01 was withdrawn from the medium. Medium was refreshed twice a week, and the apical side of the cells was washed with PBS once a week. After 14–21 days of culturing at air-exposed conditions, cells were used for further experiments. In indicated experiments, cells were treated for 48 h with IL-4 (10 ng/mL; Peprotech, Rocky Hill, NJ, USA) to increase TMEM16A expression.

To obtain nasal organoids, differentiated ALI cultures were apically washed with PBS and treated with collagenase type II (1 mg/mL; Gibco, Waltham, MA, USA), diluted in advanced DMEM/F12, at the basolateral side. The cells were incubated for 45–60 min at 37 °C until the epithelial layer detached from the transwell. Loose epithelial sheets were collected in 1 mL advanced DMEM/F12 in a 15 mL tube. The epithelial sheets were then mechanically disrupted by pipetting and subsequently strained with a 100 µM strainer. After centrifugation, epithelial fragments were resuspended in ice-cold 75% (*v*/*v*) Matrigel (diluted in airway organoid medium (Appendix A); Corning, Corning, NY, USA) and kept on ice. Then, 30 µL Matrigel droplets were plated out on a pre-warmed 24-well suspension plate. This plate was placed upside down in a tissue incubator to solidify the Matrigel droplets for 20–30 min, before adding 500 µL airway organoid medium (supplemented with FGF7 (5 ng/mL) and FGF10 (10 ng/mL)) per well.

### 4.3. Organoid Swelling Assay

For the primary screening assay, organoids were transferred to a 384-well plate, 1–3 days after organoid formation. First, organoids were harvested by dissolving the Matrigel with Cell Recovery Solution (Corning, Corning, NY, USA) during a 10 min incubation step at 4 °C. Afterwards, organoids were collected in a tube with ice-cold advanced DMEM/F12. After centrifugation, organoids were resuspended in 75% (*v*/*v*) ice-cold Matrigel (diluted in airway organoid medium) and plated out as 7 µL-droplets in a 384-well plate. Plates were centrifuged to reach the organoids at the bottom of the wells. Matrigel droplets were solidified in a tissue incubator for 10 min and subsequently 8 µL airway organoid medium supplemented with FGF7 (5 ng/mL) and FGF 10 (10 ng/mL) was added per well. Plates were covered with a breath sealing membrane (Sigma-Aldrich, St Louis, MO, USA) to prevent evaporation and incubated overnight at 37 °C. The next day, organoids were stimulated with compounds from an FDA-approved drug library (3 µM; Selleckchem, Planegg, Germany; ordered in 2016), with two compounds combined in a single well, which were mixed with a plate shaker. E_act_ (10 µM; Sigma-Aldrich, St Louis, MO, USA) was used as positive control and DMSO as negative control. Brightfield pictures were taken every 15 min for 3 h in total with a 5x objective by confocal microscopy (Zeiss LSM800) at 95% O2/5% CO_2_.

For assessment of organoid swelling in a 96-well plate format, 30 µL Matrigel droplets were scraped from the plate and transferred to tubes with ice-cold advanced DMEM/F12 to dissolve the Matrigel, 1–3 days after organoid formation. Next, organoids were centrifuged and resuspended again in ice-cold 75% (*v*/*v*) Matrigel to be plated out again in 4 µL droplets in a pre-warmed 96-well plate. The plate was placed in a tissue incubator to solidify the Matrigel droplets for 15–30 min, before adding 100 µL culturing medium (airway organoid medium supplemented with FGF7 (5 ng/mL) and FGF10 (10 ng/mL)) per well. For specified experiments, FGF7 and FGF10 were substituted by IL-4 (10 ng/mL). A swelling assay was performed 1–2 days after plating out the organoids in a 96-well plate. Organoids were imaged with fluorescence microscopy or brightfield microscopy. For fluorescence microscopy, organoids were pre-treated with calcein green AM (3 µM; Invitrogen, Waltham, MA, USA) for 30 min. Organoids were then stimulated with an agonist (forskolin (5 µM; Sigma-Aldrich, St Louis, MO, USA), ATP (100 µM; Sigma-Aldrich, St. Louis, MO, USA), E_act_ (10 µM), FDA compound (3 µM) or vehicle control) to analyze their effect on epithelial fluid secretion. Live-imaging with confocal microscopy (Zeiss LSM800) was performed to visualize the organoids at 37 °C and 95% O_2_/5% CO_2_. Pictures were taken every 15 min for 2–3 h in total with a 5× objective. Organoid swelling experiments were performed in quadruplicates. FIS experiments were performed as described before [17].

### 4.4. Analysis of Organoid Swelling Assays

Swelling of calcein green AM-labeled organoids was analyzed as described before [17]. Total organoid area per well was determined with Zen Blue image analysis software. This was used to calculate organoid surface area over time, normalized for t = 0 and with 100% as baseline. Additionally, AUC values (t = 120 min) were calculated. When indicated, baseline-corrected AUC values were calculated by subtraction of the AUC values from DMSO-treated wells from the same experimental plate.

For organoid swelling in experiments without fluorescent-labeled organoids, the OrgaQuant convolutional neural network was used to automatically recognize organoids in brightfield images, by use of the provided code [18]. Organoid surface area was estimated using OrgaQuant bounding boxes, assuming organoids had a disk shape. Particle tracking [21] was then used to follow individual organoids over 13 time points. Next, linear regression [50] was used to determine a swell rate for individual organoids and the mean swell rate of all organoids in a single well was used for further analysis. Individual organoids were excluded from analysis (1) when not recognized in minimal 8 out of 13 time points or (2) when the standard error of swell rate was >2.5 pixels/time point. Plate-normalization was performed to compare swell rates across different plates and donors by the following formula: (swell rate_well_–median swell rate_plate_)/IQR swell rate_plate_, where IQR is the inter quantile range.

### 4.5. Immunofluorescence Staining and Microscopy

Organoids plated in 30 µL droplets of Matrigel were used for immunofluorescence staining. Matrigel was dissolved by incubation with Cell Recovery Solution (Corning, Corning, NY, USA) for 15 min at 4 °C. Organoids were then fixed with 4% PFA (Aurion, Wageningen, Netherlands) for 15 min. Fixed organoids were stored in 70% EtOH or directly further processed. Organoids were embedded in pre-warmed HistoGel (Epredia, Breda, The Netherlands) and dehydrated and embedded in paraffin in a Tissue Processor (Leica). Next, 3 µM-sections were deparaffinized and antigen retrieval was performed in 10 mM citrate buffer (pH = 6; Sigma-Aldrich, St Louis, MO, USA) for 10 min. The samples were then permeabilized in 0.25% (*v*/*v*) Triton-X (Sigma-Aldrich, St Louis, MO, USA) in PBS for 10 min and subsequently blocked in 5% (*w*/*v*) BSA (Sigma-Aldrich, St Louis, MO, USA) with 0.03% Triton-X in PBS for 30 min. Next, primary antibodies (Appendix A) were incubated for 90 min and secondary antibodies (Appendix A) together with DAPI stain (1:1.000; Sigma-Aldrich, St Louis, MO, USA) for 45 min, both diluted in blocking buffer. Last, samples were mounted in Prolong Gold reagent (Thermo Fischer Scientific, Waltham, MA, USA). Images were acquired using a Leica THUNDER imager with a 40× objective. Images were processed using Las X software and ImageJ.

### 4.6. RNA Extraction, cDNA Synthesis and Quantitative Real Time PCR

RNA was extracted from nasal organoids with the RNeasy Mini-Kit (Qiagen, Venlo, Netherlands) according to the manufacturer’s protocol together with a DNA digestion step with DNase+ (Qiagen, Venlo, Netherlands). RNA yield was measured using the Qubit RNA BR assay kit (Thermo Fischer Scientific, Waltham, MA, USA). cDNA was produced with the iScript™ cDNA Synthesis Kit (Bio-Rad, Hercules, CA, USA) according to the manufacturer’s protocol. Quantitative real-time PCR (qPCR) was performed with the iQ™ SYBR^®^ Green Supermix (Bio-Rad, Hercules, CA, USA), specific primers listed in Appendix A and a CFX96 real-time detection machine (Bio-Rad, Hercules, CA, USA). Relative gene expression normalized to the housekeeping genes ATP5B and RPL13A was calculated using the software CFX Manager 3.1 (Bio-Rad, Hercules, CA, USA), according to the standard curve method.

### 4.7. Gene KO in Airway Epithelial Basal Cells Using CRISPR-Cas9

TMEM16A KO nasal epithelial cells (passage 3, n = 3 independent donors) were created using CRISPR-Cas9 technology. First, ribonucleoprotein (RNP) complexes were formed by combining multi-guide sgRNA (30 µM; Synthego, Redwood City, CA, USA), recombinant 2NLS-Cas9 nuclease (20 µM, Synthego, Redwood City, CA, USA) and optiMEM (Invitrogen, Waltham, MA, USA) supplemented with Y27632 (10 µM), followed by 10 min incubation at room temperature. Next, 1 million basal epithelial cells were made single cells with TrypLE express enzyme, and after centrifugation dissolved in optiMEM, supplemented with Y27632 (10 µM). Cells were then mixed with the RNP complexes, transferred to cuvettes and electroporated in bulk using a NEPA21 electroporator (Nepa Gene, Ichikawa City, Japan), according to previously published settings [51]. After electroporation, the polyclonal cell suspension was resuspended in basal cell expansion medium and plated out in 12-well plates. For analysis of gene editing efficiency, DNA was isolated according to the protocol of the Quick-DNA Microprep Kit (Zymo Research, Irvine, CA, USA) and DNA concentration was measured using the Qubit ds DNA BR assay kit (Thermo Fischer Scientific, Waltham, MA, USA). Regions of interest were amplified in a PCR reaction with GoTaq G2 Flexi DNA polymerase (Promega, Madison, WI, USA), and PCR-amplified samples were run on a 1,2% TBE-agarose gel for size separation. DNA fragments were excised from the gel, purified according to the Gel Extraction Kit (Qiagen, Venlo, The Netherlands) and sent for Sanger sequencing. KO efficiency was analyzed with the ICE analysis tool (www.ice.synthego.com, accessed on 21 January 2021).

### 4.8. TMEM16A Western Blot

ALI-differentiated cells were dissociated from the transwells with TrypLE express enzyme, washed twice with cold PBS and dissolved in Laemmli lysis buffer. Protein concentration was determined using the Pierce BCA Protein Assay Kit (Thermo Fischer Scientific, Waltham, MA, USA), according to the manufacturer’s protocol. Protein extracts were separated on 8% SDS-PAGE gels and transferred to a PVDF membrane (immobilon FL; Sigma-Aldrich, St Louis, MO, USA). Membranes were blocked with 1% (for TMEM16A protein) or 5% (for loading controls) (*w*/*v*) non-fat milk powder (NFM; Campina, Amersfoort, Netherlands) in Tris buffer saline with Tween-20 (Merck, Kenilworth, NJ, USA; TBS-T) for 1 h at room temperature. Primary antibodies (Appendix A) were incubated overnight at 4 °C diluted in 0.5% (*w*/*v*) NFM/TBS-T. Hsp90 was used as loading control. Secondary antibodies were incubated for 1 h at room temperature in 0.5% (*w*/*v*) NFM/TBS-T. Chemiluminescent detection was performed using SuperSignal™ West Dura Extended Duration Substrate (Thermo Fischer Scientific, Waltham, MA, USA) and the Chemidoc Touch Imaging system (Bio-Rad, Hercules, CA, USA). Quantification of band intensities was performed using ImageJ and normalized to the loading control hsp90.

### 4.9. Ussing Chamber Experiments

Nasal epithelial cells differentiated at ALI conditions for 28 days were used for Ussing chamber measurements. The day before experiments, inserts were washed with sterile PBS (Thermo Fischer Scientific, Waltham, MA, USA) for 10 min at 37 °C, 5% CO_2_. Epithelial cultures were mounted into the EasyMount Ussing Chamber System (Physiologic Instruments, Reno, NV, USA) and bathed in an HCO_3_-KRB solution, containing (in mM): 25 NaHCO_3_, 115 NaCl, 5 KCl, 1 CaCl_2_, 1 MgCl_2_, 5 D-glucose, pH 7.4. The solution was continuously gassed with 95% O_2_/5% CO_2_ and maintained at 37 °C. Monolayers were voltage-clamped to 0 mV. The transepithelial short-circuit current (I_sc_) was recorded every 10 sec using Ag–AgCl electrodes in 3M KCl agar bridges, as previously described [52], and results were normalized to an area of 1 cm^2^ and expressed as µAmp.cm^−2^ using the Acquire & Analyze software (Physiologic Instruments, Reno, NV, USA). Changes in short-circuit current (ΔI_sc_) were then calculated by averaging 5 time points before and 5 points after the addition of chemicals. Chemicals were added in the following sequence: FDA compounds (3 µM, basolateral), amiloride (amil, 10 µM, apical; Sigma-Aldrich, St Louis, MO, USA), Uridine 5′-Triphosphate trisodium salt hydrate (UTP, 0.1–100 µM, apical; Sigma-Aldrich, St Louis, MO, USA).

### 4.10. YFP-Quenching Assay

For the YFP-quenching assay in CFBE cells, CFBE parental (null CFTR) cells stably expressing halide-sensitive YFP (HS-YFP) were cultured in MEM 1× (Corning, Corning, NY, USA) and 1 mg/mL of Hygromycin B (Sigma-Aldrich, St Louis, MO, USA). Cells were seeded 50.000 cells/well on clear-bottom 96-well black microplates suitable for high-content imaging (90 μL per well). Forty-eight hours after plating, cells were washed twice and incubated for 25 min at 37 °C with 65 μL of standard PBS (in mM: 137 NaCl, 2.7 KCl, 8.1 Na_2_HPO_4_, 1.5 KH_2_PO_4_, 1 CaCl_2_, 0.5 MgCl_2_, pH 7.4) containing vehicle alone (0.1% (*v*/*v*) DMSO) or with different FDA compounds (3 or 10 μM). After 25 min, the plate was transferred to a microplate reader (Tecan, Mannedorf, Switzerland). Each assay consisted of a continuous 12 s fluorescence reading—2 s before and 10 s after injection of 170 μL of iodide-rich PBS (in mM: 137 KI, 2.7 KCl, 8.1 Na_2_HPO_4_, 1.5 KH_2_PO_4_, 1 CaCl_2_, 0.5 MgCl_2,_ pH 7.4) containing 1 μM of Ionomycin (Sigma-Aldrich, St Louis, MO, USA). Each well was normalized to their own initial fluorescence and linear fits were performed for each point. The fluorescence quenching rate (QR) represents the steepest slope within the different slopes previously calculated.

For the YFP-quenching assay in HT-29 cells, stably expressing the iodide-sensitive enhanced yellow fluorescent protein (eYFP-I152L), cells were plated in transparent 96-well plates. After 24 h of culturing to 80–90% confluence, they were incubated with or without FDA compounds (3 μM) in a gluconate-substituted Ringer solution (in mM: NaCl 100, Na-Gluconate 40, KCl 5, MgCl_2_ · 6 H_2_O 1, CaCl_2_ · 2 H_2_O 2, Glucose 10, HEPES 10). Iodide was added as a symmetrical iodide-substituted Ringer solution (in mM: NaCl 100, NaI 40, KCl 5, MgCl_2_ · 6 H_2_O 1, CaCl_2_ · 2 H_2_O 2, D-Glucose 10, HEPES 10) and 5 µM ATP was then added acutely in the final 1:1 mixed Ringer solution (NaCl 100, Na-Gluconate 20, NaI 20, KCl 5, MgCl_2_ · 6 H_2_O 1, CaCl_2_ · 2 H_2_O 2, D-Glucose 10, HEPES 10). The final iodide concentration on each well was 20 mM for every experiment. Total intracellular YFP-fluorescence intensity in each well was measured continuously with a fluorescence microplate reader (NOVOstar, BMG Labtech, Ortenberg, Germany) kept at 37 °C, using an excitation wavelength of 485 nm and emission detection at 520 nm. Background fluorescence was subtracted and data were normalized to the initial fluorescence. The initial rate of maximal fluorescence decay caused by iodide influx was then calculated as a measure of anion conductance.

### 4.11. Statistical Analysis

For organoid swelling assays, four technical replicates were used per experimental condition. All results are shown as mean values ± SD from biological replicates, unless indicated otherwise. For statistical analyses of differences, an (un)paired *t*-test or one/two-way ANOVA with indicated post hoc test were used, as indicated in the figure legends. *p*-values < 0.05 were considered as statistically significant. Statistical analyses were performed using Graphpad Prism 9 or R v.4.0.3.

## Figures and Tables

**Figure 1 ijms-23-12657-f001:**
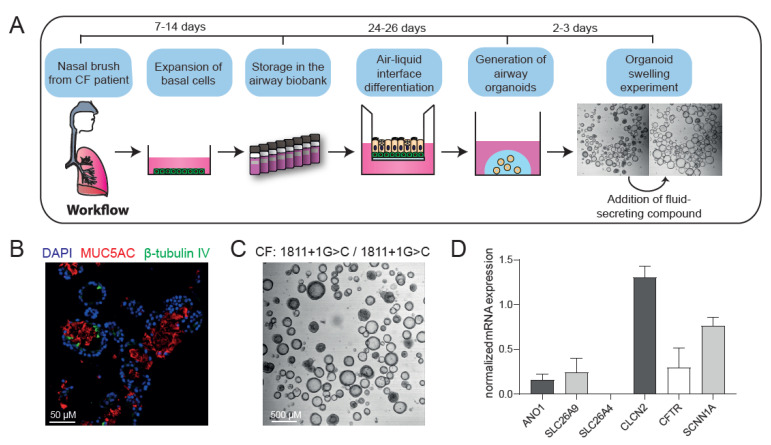
Characterization of CF nasal organoids and CFTR-independent organoid swelling. (**A**) Schematic representation of the project workflow from nasal brush towards organoid swelling experiments; (**B**) immunofluorescence staining of nasal organoids with the secretory cell marker MUC5AC (red), ciliated cell marker β-tubulin IV (green) and DAPI (blue) from a CF donor (F508del/S1251N); (**C**) brightfield image showing intrinsic lumen formation of unstimulated nasal organoids from a CFTR-null donor (1811+1G>C/1811+1G>C); (**D**) mRNA expression in CFTR-null nasal organoids of the following ion channels/transporters: ANO1 (TMEM16A), SLC26A9, SLC26A4, CLCN2, SCNN1A and CFTR (n = 3 independent donors; W1282X/1717-1G>A, R553X/R553X, G542X/CFTRdele2.3 (21 kb)); (**E**) confocal images of CFTR-null (G542X/CFTRdele2.3(21 kb)) nasal organoids, stimulated with forskolin (5 µM), ATP (100 µM) or Eact (10 µM) at 0 and 120 min; (**F**) quantification of CFTR-null (G542X/CFTRdele2.3(21 kb), n = 5 replicates) nasal organoid swelling after stimulation with forskolin, ATP or Eact; (**G**) area under the curve (AUC) plots of nasal organoid swelling in three CFTR-null donors (n = 3 independent donors; W1282X/1717-1G>A, R553X/R553X, G542X/CFTRdele2.3 (21 kb); 2–6 replicates per donor) after stimulation with forskolin, ATP or Eact. Analysis of difference with control was determined with a one-way ANOVA with Dunnett’s post hoc test (**G**). *** *p* < 0.001, **** *p* < 0.0001.

**Figure 2 ijms-23-12657-f002:**
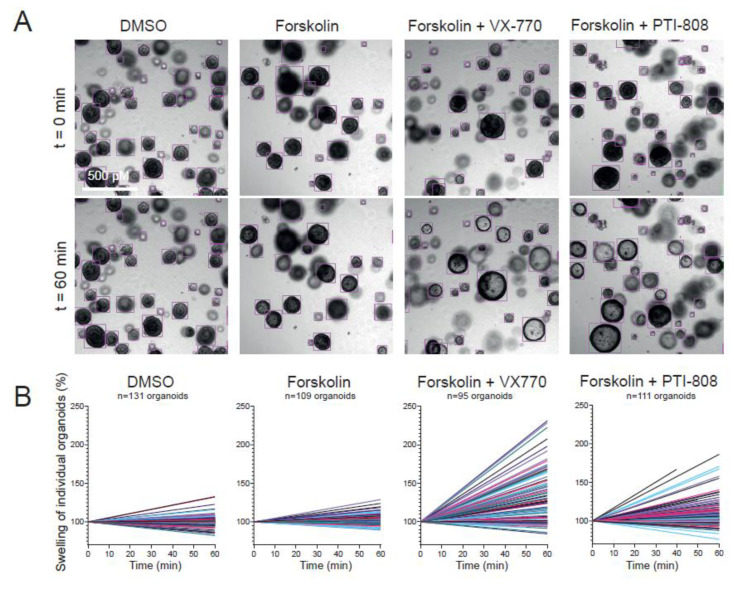
Quantification of nasal organoid swelling in a 384-well plate format by the OrgaQuant neural network. (**A**) Representative brightfield images showing automatic recognition of CF nasal organoids (F508del/S1251N) with the OrgaQuant model [18]. To examine swelling, organoids were treated with the vehicle DMSO, forskolin alone (5 µM) or a combination of forskolin (5 µM) with the CFTR potentiators VX-770 (5 µM) or PTI-808 (1 µM); (**B**) graphs show percentage change in surface area relative to t = 0 (100%) of individual organoids, treated with vehicle DMSO, forskolin, forskolin with VX-770 or forskolin with PTI-808, corresponding to the wells from (**A**). Each line represents an individual organoid; (**C**) swell rates (pixels/time point) of individual organoids from the wells shown in (**A**) are displayed (one representative well per condition is shown); (**D**) analysis of FIS using the conventional quantification method in fluorescent-labeled organoids, using the same donor as in (**A**–**C**) (n = 1 donor, F508del/S1251N, 2 biological replicates). Organoids were treated with vehicle DMSO, forskolin (5 µM), forskolin with Vx770 (5 µM) or forskolin with PTI-808 (1 µM). AUC is used as outcome measurement for organoid swelling; (**E**) correlation between two analysis methods for FIS in a similar donor: in the new developed method, organoids in brightfield images are recognized using OrgaQuant and swell rate (pixels/time point) is used as outcome measurement for swelling. In the conventional method, fluorescent-labeled organoids are recognized with image software Zen Blue and AUC values are used as outcome measurement for swelling; (**F**) swell rates of individual organoids within a single well stimulated with DMSO or E_act_ (10 µM) in a CFTR-null donor (1811+1G>C/1811+1G>C). Analysis of differences was performed using unpaired *t*-tests (**F**), one-way ANOVA wit Tukey post hoc test (**C**,**D**) or Pearson correlation (**E**). ns = non-significant, * *p* < 0.05, ** *p* < 0.01, *** *p* < 0.001, **** *p* < 0.0001.

**Figure 3 ijms-23-12657-f003:**
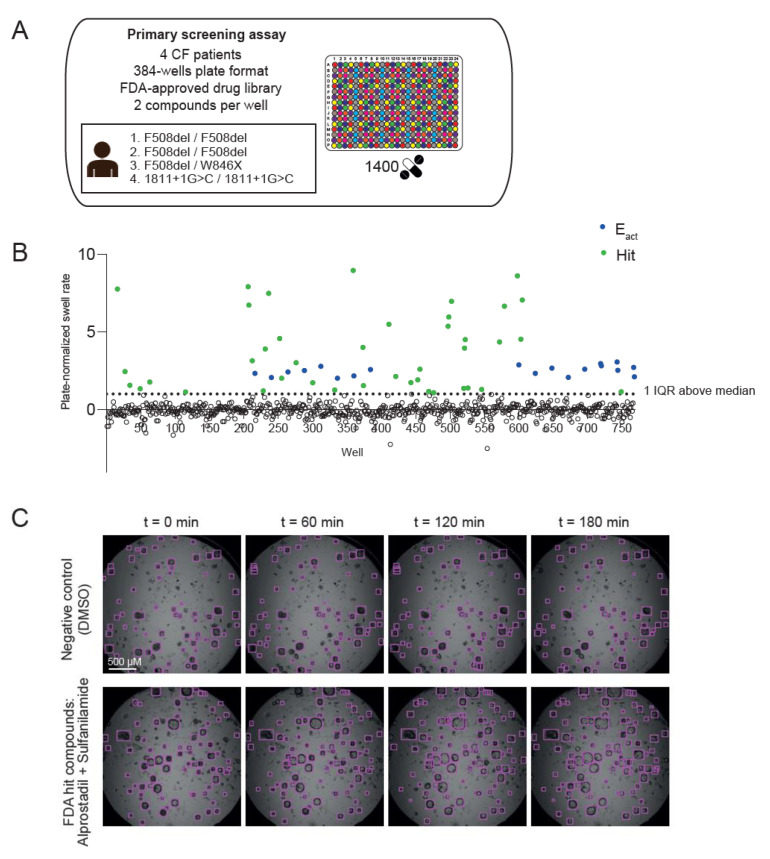
Primary screening assay of an FDA-approved drug library in CF nasal organoids. (**A**) ~1400 FDA-approved drugs (3 µM) were screened in a 384-well plate format in nasal organoids from 4 pwCF. Two compounds were combined in a single well; (**B**) the graph represents mean plate-normalized swell rates of four CF donors. A total of 90 compounds (shown in green), divided over 45 wells, with a plate-normalized swell rate above 1 IQR above the median were selected from the primary screening assay for the secondary screening assay. E_act_ (10 µM, shown in blue) was used as positive control (n = 4 independent donors; F508del/F508del, F508del/F508del, F508del/W846X, 1811G+1>C/1811+1G>C, 1–3 replicates per donor); (**C**) representative brightfield images showing automatic recognition of nasal organoids (CF: 1811+1G>C/1811+1G>C) using the OrgaQuant model [18]. Examples are shown from a well containing DMSO as negative control (upper panel) and a well containing FDA hit compounds (lower panel); (**D**) graphs show percentage change in surface area relative to t = 0 (100%) of individual organoids, treated with vehicle DMSO (left panel) or FDA hit compounds (right panel), corresponding to the organoids shown in (**C**). Each line represents an individual organoid; (**E**) quantification of swell rates of individual organoids from the example wells shown in (**C**,**D**). Analysis of differences was performed using an unpaired *t*-test (**E**). **** *p* < 0.0001.

**Figure 4 ijms-23-12657-f004:**
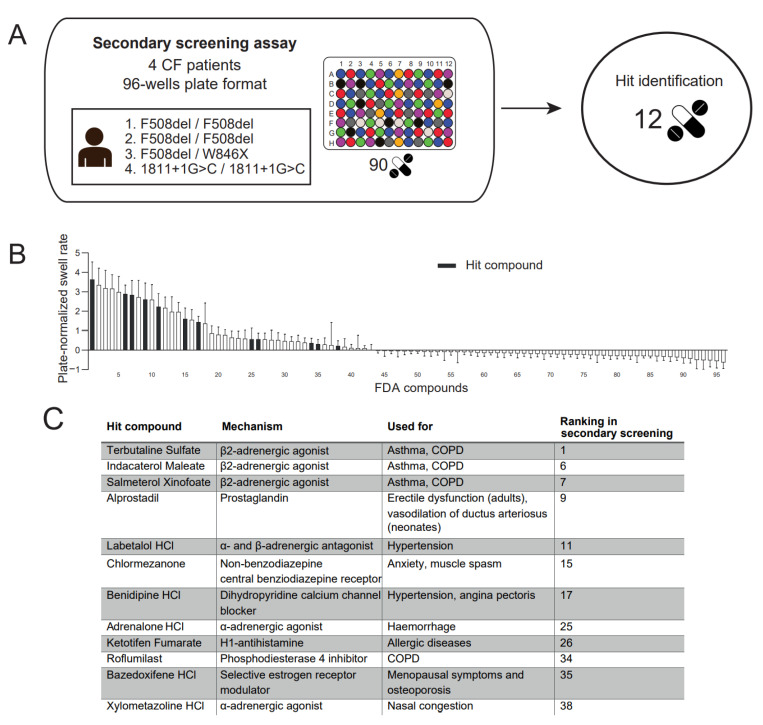
Secondary screening assay and validation of FDA hit compounds in CFTR-null donors. (**A**) A total of 90 hit compounds identified in the primary screening assay were further validated in the conventional 96-well plate format with one compound per well; (**B**) the graph represents mean plate-normalized swell rates of four donors. Hit compounds were selected based on swell rate and working mechanism (n = 4 independent donors; F508del/F508del, F508del/F508del, F508del/W846X, 1811G+1>C/1811+1G>C, 3 replicates per donor); (**C**) overview of the 12 hit compounds with their working mechanism, disease application and ranking in the secondary screening assay; (**D**) the 12 hit compounds were further evaluated in an organoid swelling assay with CFTR-null nasal organoids (n = 3 independent donors: G542X/CFTRdele2.3(21kb), W1282X/1717-1G>A, R553X/R553X, n = 2–7 measurements per donor). The compounds are ranked based on their effect size in the secondary screening assay, shown in (**B**). The conventional image analysis was applied using fluorescent-labeled organoids. Organoid swelling is shown as AUC values from measurements of 120 min; (**E**) representative confocal images of CFTR-null (R553X/R553X) nasal organoids, stimulated with Terbutaline Sulfate or Alprostadil (both 3 µM) as example of two hit compounds at 0 and 120 min. (**F**) Quantification of CFTR-null (R553X/R553X) nasal organoid swelling after stimulation with Terbutaline Sulfate or Alprostadil (both 3 µM). Differences with baseline are analyzed using a one-way ANOVA with Dunnett’s post hoc test (**D**). ns = non-significant, ** *p* < 0.01, *** *p* < 0.001, **** *p* < 0.0001.

**Figure 5 ijms-23-12657-f005:**
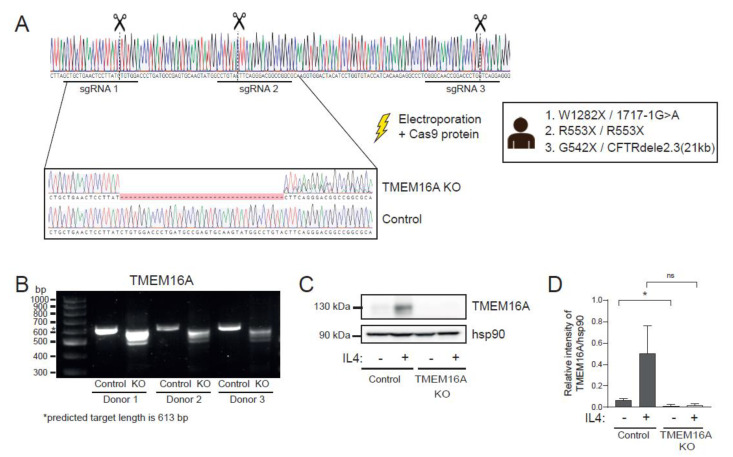
Generation and validation of TMEM16A KO nasal epithelial cells. CRISPR-Cas9 based gene editing was used to generate a TMEM16A KO in CFTR-null nasal cells. (**A**) Graphical overview showing binding sites of three sgRNA molecules and Sanger sequencing traces after electroporation; (**B**) DNA gel showing the PCR-amplified products of the targeted *TMEM16A* locus for KO and control samples of 3 CFTR-null donors (G542X/CFTRdele2.3 (21 kb), W1282X/1717-1G>A, R553X/R553X). Predicted length of the PCR product was 613 bp; (**C**) representative Western blot for TMEM16A protein of ALI-differentiated TMEM16A KO and control cells. To increase TMEM16A expression, some cells were treated with IL-4 for 48 h; (**D**) quantified band intensity of TMEM16A protein in Western blots (n = 3 independent donors); (**E**) functional validation of ALI-differentiated TMEM16A KO cells with Ussing chamber measurements. TMEM16A activity was determined based on Ani9-sensitive (1 µM) UTP-induced (100 µM) currents. Representative traces are shown of one donor and (**F**) UTP-induced currents were quantified for all donors, with and without Ani9-treatment (n = 3 independent donors). All cells were treated with amiloride and indicated cells were treated with IL-4 for 48 h. Analysis of differences was performed using paired *t*-tests (**D**) or a 2-way ANOVA with Tukey post hoc test (**F**). ns = non-significant, * *p* < 0.05, *** *p* < 0.001, **** *p* < 0.0001.

**Figure 6 ijms-23-12657-f006:**
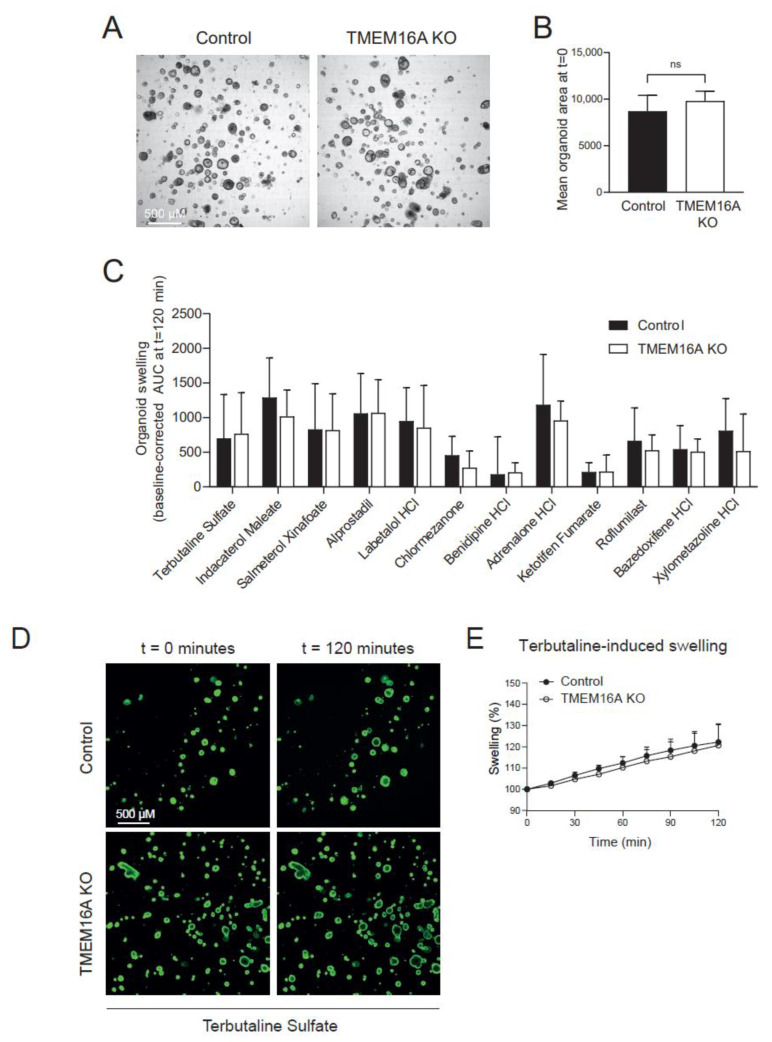
Hit compound validation in TMEM16A KO nasal organoids. (**A**) Brightfield images showing intrinsic lumen formation, without any stimulation, in both control and TMEM16A KO nasal organoids (G542X/CFTRdele2.3(21kb)); (**B**) quantification of organoid lumen size in control and knockout organoids (n = 3 independent donors); (**C**) validation of hit compounds on nasal organoid swelling in TMEM16A KO and control organoids (n = 3 independent donors, 2–6 measurements per donor); (**D**) representative confocal images (G542X/CFTRdele2.3 (21 kb)) of TMEM16A KO and control nasal organoids, stimulated with Terbutaline Sulfate (3 µM) as example of one of the hit compounds; (**E**) quantification of nasal organoid swelling after stimulation with Terbutaline Sulfate (3 µM) in TMEM16A KO and control organoids (n = 3 independent donors). Analysis of difference was performed with a paired (**B**) or unpaired (**C**) *t*-test. No significant results were found. Ns = non-significant.

**Figure 7 ijms-23-12657-f007:**
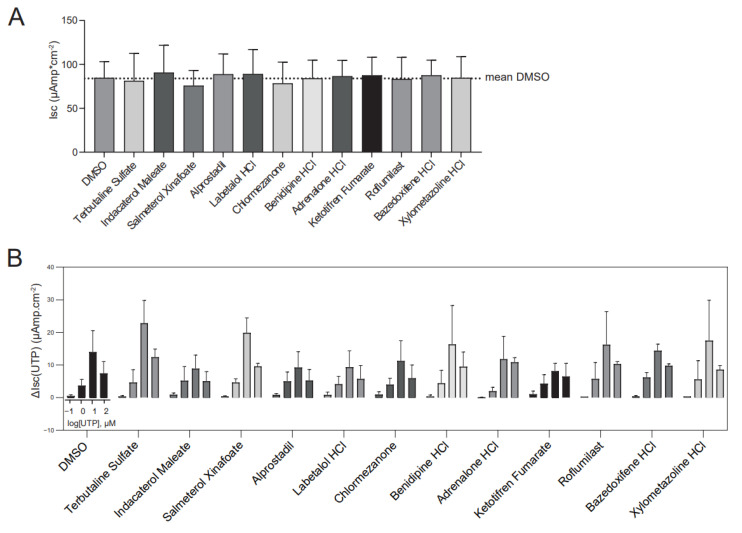
Effect of the hit compounds on TMEM16A in other in vitro model systems. (**A**) The effect of the hit compounds (3 µM) on chloride conductance was determined in Ussing chamber measurements with ALI-differentiated CFTR-null nasal cells (n = 3 independent donors; each compound was measured in at least 2 different donors; n = 1–6 measurements per donor). No significant differences were found between one of the compounds and DMSO; (**B**) assessment whether the hit compounds (3 µM) enhance UTP-induced currents in Ussing chamber measurements. Experiments were conducted by stimulating different concentrations of UTP with hit compounds in ALI-differentiated CFTR-null nasal cells (n = 3 independent donors, each compound was measured in at least 2 different donors; n = 2–3 measurements per donor). No significant results were found between one of the compounds and DMSO, for any concentration of UTP; (**C**,**D**) effect of the 12 hit compounds (3 µM) on ionomycin-induced (1 µM) iodide influx was assessed with an YFP-quenching assay in CFBE cells. TMEM16A-dependency was demonstrated with sensitivity for Ani9 (3 and 10 uM, shown in red). DMSO was used as negative control (shown in green) and E_act_ (3 and 10 µM) as positive control (shown in blue). For quantification, quenching rates were normalized to the control; (**E**,**F**) effect of a selection of 6 hit compounds (3 µM) on ATP-induced (5 µM) iodide influx was analyzed in HT-29-YFP cells. TMEM16A-dependency was demonstrated with sensitivity for Ani9 (10 uM, shown in red) and DMSO was used as negative control (shown in green). For quantification, quenching rates were normalized to the control. Analysis of differences were performed with one-way ANOVA and Dunnett’s post hoc test (**A**,**D**,**F**) or a two-way ANOVA with Dunnett’s post hoc test (**B**). ** *p* < 0.01, **** *p* < 0.0001.

## Data Availability

All data are provided with the manuscript. Code is available at https://github.com/UMCU-BeekmanLab/OrgaQuantBeekman.

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
