# Peer review of "Drug Repurposing for Cystic Fibrosis: Identification of Drugs That Induce CFTR-Independent Fluid Secretion in Nasal Organoids"

_ijms, 2022, doi:10.3390/ijms232012657_

Round 1

Reviewer 1 Report

General comments:

The manuscript by Rodenburg et al. reports a very interesting study proposing a new methodology to establish automated screenings for CFTR-independent fluid secretion in patient-derived samples. The method employs organoids derived from primary nasal cells collected from CF patients and an AI-assisted open-source algorithm that tracks and quantifies organoid swelling in brightfield images. The authors used this method to screen a library of 1400 FDA-approved drugs and identified 90 hits from which they selected 12 for further study in CFTR-null organoids by conventional fluorescent FIS analysis. The authors proceeded by investigating whether the alternative chloride channel TMEM16A was involved in the observed CFTR-independent swelling effects, by using CRISPR-Cas9 to generate TMEM16A KO nasal cells and respective organoids. They concluded that TMEM16A was not mediating the CFTR-independent swelling in nasal organoids. Notably, the authors observed that none of the 12 hit drugs changed chloride conductance in nasal, bronchial or intestinal monolayers, as measured by Ussing chamber or by quenching of an halide-sensitive YFP sensor.

The manuscript is well structured and the results clearly described. However, there are a few points that require further clarification by the authors.

Specific comments:

The goal of identifying new drugs that can improve epithelial fluid secretion independently of CFTR is critical, not only for the nearly 1/5 of CF patients that are not eligible for CFTR modulator therapy, but also for those patients that simply cannot afford them. The possibility of using patient-derived organoids in drug screenings constitutes a remarkable advance in improving the chances of rapidly finding effective drugs. Moreover, the authors’ rationale of starting the search for CFTR-independent CF modulators among FDA-approved drugs is highly relevant.

What is unclear is the reason why the authors chose not to do the initial screenings in the available CFTR-null organoids? How can the authors ensure that the detected effects were CFTR-independent? Wouldn’t it be more reasonable to concomitantly use CFTR inhibitors (at least in the secondary screen) to exclude potential confounding effects of the drugs on the trafficking or function of mutant CFTR? Were the chosen compounds the most efficient in inducing CFTR-independent swelling or the stronger effects were also influenced by CFTR-residual function?

Conversely, the authors argue towards the clinical translation of the identified hit compounds, namely b2-agonists. Since many promising CF modulator compounds have failed to translate clinically, it would be important to determine how these compounds compare to VX-drugs regarding the kinetics and magnitude of FIS response in F508del/F508del organoids. Are these additive with VX-drugs? Can the author provide these data and extrapolate/discuss the potential therapeutic gain of using these compounds?

Have the authors tested for any additive effects between compounds from different MoA categories? On this note, the authors have given a particular emphasis to b-agonists in their final hit shortlist. It would be important for them to discuss their observation that labetalol, a a- and b-adrenergic antagonist, induces swelling to an extent comparable to that of b-(namely terbutaline) and, for that matter, a-(adrenalone) agonists. Were there other VGCC blockers among the 90 initial hits?

Still regarding the data on Fig.4D, the effect of ketodifen appears to be equivalent, or even lower, than that of Benidipine. However, whereas the latter is marked as not significant, ketodifed's effect is marked with a p-value <0.001. How do the authors justify this discrepancy? A similar argument could be made for chlormezanone.

In addition, how many of the 12 compounds analyzed in Fig 4D are among the 12 top ranking compounds in Fig 4B? The magnitude of their effect in conventional FIS appears much lower than the normalized rates in Fig 4B; whereas in Fig 2 the authors showed conventional FIS to be more sensitive.

Not diminishing the relevance of demonstrating the feasibility of the KO strategy in nasal organoids, it is also unclear why, among the many candidate channels described in the manuscript, the authors chose TMRM16A as the putative channel mediating the observed hit-induced swelling effects. They show the channel to be barely expressed in these cells (without IL-4 stimulation) showing minimal activity in the conditions used for screening and subsequent validations. Why not CLCN2, which they show to be much more expressed and is also activated by cAMP?

Did the authors test the hit-compounds together with IL-4?

Minor points:

Statistical analysis of data in Fig.1G requires one way ANOVA followed by Dunnett’s posthoc tests.

Given IJMS’s broad readership, the authors should provide a short description of the molecular/functional dysfunctions caused by each of the CFTR mutations presented in the manuscript.

Some of the figures in the pdf are at very low resolution. Captions are cut in some of panels. Please confirm the quality of the original images.

Reviewer 2 Report

Finding alternative drugs for people with CF who are not eligible for CFTR modulator therapy is a main concern in the CF field. One way to identify quickly such new compounds is to screen FDA-approved drugs as modulators of non-CFTR dependent epithelial fluid secretion because most of the non-eligible people bears CFTR mutations which lead to an absence of CFTR protein. This paper reports a smart and technically sound strategy to identify such drugs by combining various model of primary nasal cells genetically modified and functional assessment of ion and fluid secretion.

Methods are sound and well-designed.

Major points

The following points should be clarified:

In this present version, the paper is written to expert in the CF field. It must be adapted to a broader audience.

Results section:

Classification of the CFTR mutations is mentioned in the title of paragraph 2.1 but class I/VII is not explained. Either the classification should be briefly explained or a brief description of the resulting effect of the chosen mutations should be given, i.e. mutations given rise to no CFTR mRNA/protein.

Some compound should also be defined such as Eact or at least defined their role in the experiment (i.e activation of TMEM16A).

The level of mRNA expression in CFTR-null nasal organoids of selected ion channels/transporters should be further discussed as very different level of expression are observed. The level of ANO1, SLC26A9 and CFTR is low as expected for CFTR, whereas CLCN2 is highly expressed. These data should be argued. A similar experiment with wt-CFTR nasal organoids should be shown to this aim.

The mRNA level of each ion channels/transporters tested should be shown in each donor as it’s interesting to draw a comparison with the level of CFTR expressed.

Same comment for the AUC of plots of the three donors as shown in Figure 1G.

Why the CFTR-null donor (1811+1G>C/1811+1G>C) organoids are not used to assess mRNA expression or swelling?

The term potentiator is well-known in the CF field but should again be briefly explained.

Quality of some figures should be improved in the main text (better in supplemental figures). The representation of the data should be homogenized (Fig. 1G, Fig. 4D: mean DMSO). When error bars are huge, an alternative representation of the data would be more appropriate to understand the spread if data.

Materials and Methods section:

Well documented and very informative with many data in the supplementary table some of which existing in the previous methodological paper of the team but with additional material (TMEM KO) ...

Discussion section:

Do you really think that creating KO cells of each candidate ion channels/transporters is a sustainable strategy? This point needs to be further discussed.

Minor text correction:

Line 443: penicillin instead of pencillin.

Round 2

Reviewer 1 Report

The authors have answered adequately to my comments.

Reviewer 2 Report

None